# Assessing the Influence of Fumigation and *Bacillus Subtilis*-Based Biofungicide on the Microbiome of Chrysanthemum Rhizosphere

**Huijie  Chen, Jiamiao Zhao, Jing Jiang, Sumei Chen, Zhiyong Guan, Fadi Chen, Weimin Fang and Shuang Zhao \***

State Key Laboratory of Crop Genetics and Germplasm Enhancement, Key Laboratory of Landscaping, Ministry of Agriculture and Rural Affairs, College of Horticulture, Nanjing Agricultural University, Nanjing 210095, China; 2016204031@njau.edu.cn (H.C.); 2017104106@njau.edu.cn (J.Z.); 2017804128@njau.edu.cn (J.J.); chensm@njau.edu.cn (S.C.); guanzhy@njau.edu.cn (Z.G.); chenfd@njau.edu.cn (F.C.); fangwm@njau.edu.cn (W.F.)

\* Correspondence: zhaoshuang@njau.edu.cn; Tel.: +86-025-8439-6502

**Abstract:** Chrysanthemum is an important ornamental species in China.  However, sustained monoculture often leads to a decline in soil quality, in particular to the build-up of pathogens. *Fusarium* wilt, a severe disease in chrysanthemum monoculture systems, was effectively controlled by fumigation and/or the application of a biofungicide in our previous study.  However, the mechanisms underlying disease suppression remain elusive.  Here, a series of greenhouse experiments were conducted to characterize the effect on the chrysanthemum rhizosphere microbiome of the fumigant dazomet (DZ) and of a biofungicide based on *Bacillus subtilis* NCD-2 (BF). The results indicated that the BF treatment increased bacterial diversity by 4.2%, while decreasing fungal diversity by 21.3%. After two seasons of BF treatment, the abundance of microbes associated with disease suppression such as *Bacillus* spp. and *Trichoderma* spp. increased 15.1-fold and 4.25-fold more than that of the control, while the pathogenic *Fusarium oxysporum* was decreased by 79.20% when compared to the control.  Besides, the DZ treatment reduced both bacterial and fungal diversity 7.97% and 2.73% respectively, when compared with the control.  The DZ treatment controlled *Fusarium* wilt disease and decreased the abundance of *F. oxysporum* in the first year, but the abundance of the *F. oxysporum* was 43.8% higher after two years in treated soil than in non-treated soil. Therefore, the application of BF has a great potential for the control of *Fusarium* wilt disease in chrysanthemum by changing soil microbiome structure and function.

**Keywords:** microbial community; microbial diversity; fumigation; biofungicide; chrysanthemum

## 1. Introduction

Chrysanthemum (*Chrysanthemum morifolium*) is a highly prized and potentially very profitable ornamental species [1].  Its production can be severely compromised by the presence in the soil of *Fusarium oxysporum*, the causative organism of *Fusarium* wilt [2].  This disease is particularly difficult to control because the pathogen's chlamydospores can survive over a long period in the soil, while the fungus can attack a wide range of other plant species [3].

In the past, growers have resorted to fumigating the soil with methyl bromide to control *Fusarium* wilt, but this practice was abandoned due to its damaging impact on the atmosphere's ozone layer [4]. As the use of methyl-bromide and its derivatives continue to be phased out, several alternatives such as 1, 3-dichloropropene (1, 3-D), chloropicrin (CP), dazomet (DZ), and dimethyl disulfide (DMDS) are

being used increasingly [5]. Among them, the dazomet has been used frequently to control soil-borne diseases in plant cultivation because it is versatile, highly effective, and relatively easy to use [6,7].

With the discovery of pathogen antagonists, efforts to develop biological control mechanisms have become more and more popular. *Paenibacillus polymyxa* [8], *Trichoderma harzianum* [9], and *Bacillus subtilis* [10], were widely reported antagonistic microbes shown to have the potential ability to suppress *Fusarium* wilt of various plants. Furthermore, soil suppression of disease induced by biofungicides have been widely described [11,12] and are more frequently related to the soil microbiome [3].

The soil microbiome has a major role in maintaining soil health [13]. A decrease in its diversity can encourage the development of a number of soil-borne plant diseases [14]. While the effectiveness of soil fumigation and the use of biofungicides in the context of controlling chrysanthemum *Fusarium* wilt disease is well accepted [7], the effect of these treatments on the composition of the soil microbiome has to date been largely overlooked.

In our previous study, chrysanthemum *Fusarium* wilt disease was efficiently controlled by consecutive applications of biofungicide (BF) and soil fumigation [15], and the effects of BF and soil fumigation application on the composition of rhizosphere microbial communities were evaluated by DGGE (Denaturing Gradient Gel Electrophoresis) fingerprinting of the 16S rRNA gene and ITS (Internal Transcribed Spacer) gene. The results suggested that both BF and soil fumigation application could reshape the composition of the rhizosphere microbial communities. However, due to methodological limitations, we only focused on a small fraction of the soil microbes.

In this study, we expanded upon the previous study by using Illumina high-throughput sequencing of the bacterial 16S rRNA gene and fungal ITS region to characterize the impact on the soil microbiome associated with the roots of chrysanthemum of either DZ fumigation or the use of a *B. subtilis*-based biofungicide. The results of this study can provide guidance for the control of *Fusarium* wilt of chrysanthemum, improve the knowledge of the composition of microbial communities in the rhizosphere soil, and lead to a better understanding of microbe roles in the soil after Dazomet fumigation and *Bacillus subtilis* enhanced bio-fungicide application.

## 2. Materials and Methods

### 2.1. Experimental Site

The experiment was conducted between May and October in both 2015 and 2016 at the Nanjing Agricultural University's Chrysanthemum Germplasm Resources Conservation Center (Nanjing, China). Prior to the experiment, the field had a five-year history of continuous chrysanthemum monoculture and suffered from severe *Fusarium* wilt. Prior to the initiation of the experiment, the soil was sandy loam, had a pH of 6.96, and an EC (electric conductivity) of 467.67 $\mu S \cdot cm^{-1}$, and contained 11.60 g organic matter$\cdot kg^{-1}$, 0.09 g available N$\cdot kg^{-1}$, 0.36 g available K$\cdot kg^{-1}$, and 0.18 g available P$\cdot kg^{-1}$. Young plants of the chrysanthemum cultivar '*Jinba*' (provided by Honghua Horticulture Co. Ltd., Shanghai, China) were established by growing cuttings in perlite for three weeks in a greenhouse which delivered a 16 h photoperiod and a day/night temperature regime of 28 °C/22 °C.

### 2.2. Soil Fumigant and Biofungicide

Dazomet (3,5-dimethyl-1,3,5-thiadiazinane-2-thione, DZ, pure ≥95.0%) can release methyl isothiocyanate, which is often used to treat soil before chrysanthemum replanting and the granular preparation of DZ was purchased from Nantong Shizhuang Chemical Co., Ltd. (Nantong, Jiangsu, China). For the BF treatment, the biofungicide 'Xinzhinong' wettable powder, which contains $10^9$ spores$\cdot g^{-1}$ of *B.subtilis* NCD-2, was purchased from Baoding Kelvfeng Biochemical Technology Co., Ltd. (Baoding, Hebei, China). The NCD-2 was cultured in liquid King's medium B at 24 ± 1 °C for 24 h with continuous shaking (150 rpm) and harvested by centrifugation (6520 g, 20 min) with the resulting pellet suspended in 0.1 M MgSO$_4$ in a ratio of 1 mg pellet per 1 mL. The suspension was mixed with 10% (v/v) glycerol and then with an equal volume of autoclaved 1.5% (w/v) sodium alginate, after

which the wetting agent calcium lignosulfonate was added (7%, w/w). The resulting preparation was spread thinly over a glass plate and allowed to air dry in a laminar air flow cabinet at 24 °C for 1 h to form a powder containing about 15% water.

## 2.3. Experimental Design

The nine experimental plots were set out as a randomized complete block with three replicates and three treatments. The experiment was performed in 2015 and repeated in 2016. The treatments comprised (1) control (non-treated soil), (2) chemical fumigation (DZ) (30 g dazomet m$^{-2}$, applied by mixing the microgranules into the top soil (water content: 70%); the soil surface was then covered with a 0.08 mm thick plastic film for 20 days, and then left for a further seven days prior to planting), and (3) biofungicide (BF) (30 g biofungicide m$^{-2}$ applied by mixing the powder into the top soil). Before planting, the soil was plowed to a depth of 20 cm. Each plot was measured at 80 cm × 40 cm and was planted with 24 plants.

## 2.4. Soil Sampling and DNA Extraction

In each year, soil clinging to chrysanthemum plants was collected (18 in total, three for each treatment) from plants up-rooted 90 days after transplantation. The soil samples were stored at –80 °C until required for analysis. DNA was extracted from 250 mg aliquots of the rhizosphere soil using a power soil DNA isolation kit (MoBio laboratories, Carlsbad, CA, USA), following the manufacturer's protocol. The concentration and quality of the resulting DNA preparations were determined spectrometrically.

## 2.5. PCR Amplification and Illumina Sequencing

Two pairs of primers were used to amplify the DNA extracted from the rhizosphere soil samples. One targeted the V4 hypervariable region of the 16S rRNA gene (515F: 5'-GTGCCAGCMGCCGCGGTAA, 806R: 5'-GCACTACHVGGGTWTCTAAT), and the other the internal transcribed spacer of fungal rRNA gene (ITS5-1737F: 5'-GGAAGTAAAAGTCGTAACAAGG, ITS2-2043R (5'-GCTGCGTTCTTCATCGATGC). PCRs were conducted with Phusion$^{®}$ High-Fidelity PCR Master Mix (New England Biolabs. Ipswich, MA, USA) using the following PCR program: initial denaturation at 95 °C for 3 min; 35 cycles of denaturation at 94 °C for 30 s, primer annealing at 50 °C for 1 min, extension at 72 °C for 1 min, and a final extension of 10 min at 72 °C. Then, the amplicons were separated electrophoretically through a 2% agarose gel. Only those amplicons producing strong bands at 200–300 bp were retained. The pair of amplicons derived from each soil sample was mixed in equidensity ratios and then purified using a gel extraction kit (Qiagen, Hilden, Germany). Sequencing libraries were generated using a TruSeq$^{®}$ DNA PCR-Free Sample Preparation kit (Illumina, San Diego, CA, USA) and index codes were added. The quality of the library was assessed using a Qubit 2.0 fluorometer and a bioanalyzer 2100 device (Agilent Technologies, Santa Clara, CA, USA). The library was sequenced using a HiSeq2500 device (Illumina) and 250 bp paired-end reads were generated by Novogene Biotechnology Inc. (Beijing, China).

Raw data containing adapters or low-quality reads would affect the assembly and analysis. Thus, to get high-quality clean reads, raw reads were further filtered using FASTP. Paired end clean reads were merged as raw tags using FLSAH (version 1.2.11) [16], and noisy sequences of raw tags were filtered by QIIME (version 1.9.1) [17] pipeline under specific filtering conditions to obtain the high-quality clean tags. Clean tags were searched against the reference database to perform reference-based chimera checking using the UCHIME algorithm. All chimeric tags were removed and finally, the obtained effective tags were used for further analysis. The effective tags were clustered into operational taxonomic units (OTUs) of ≥97% similarity using the UPARSE [18] pipeline. We chose a representative sequence from each OTU, and the Ribosomal Database Project (RDP) classifier (the RDP bacterial 16S database for 16S rRNA data and the UNITE fungal ITS database for ITS data) was used to assign

taxonomic information [19,20]. The MOTHUR (version 1.25.1) standard operating procedure (SOP) was employed for further analyses.

## 2.6. Statistical and Bioinformatic Analyses

A one-way analysis of variance (ANOVA) was used to identify where treatment means differed significantly ($p < 0.05$) from one another, and Duncan's multiple range test was used to compare sets of means. The necessary computations were carried out using routines implemented in Microsoft Excel 2017 and SPSS v20.0 software (SPSS, Chicago, IL, USA). Alpha diversity (including Chao1, Shannon, Faith's phylogenetic diversity, evenness) and beta diversity on both weighted and unweighted unifrac were calculated using QIIME v1.7.0 software and R software (version 2.15.3). Non-metric multi-dimensional scaling analysis was also implemented using routines included in R (version 2.15.3) software.

## 3. Results

### 3.1. Subsection Illumina Sequencing and Operational Taxonomic Unit (OTU) Classification

After trimming and quality control, the raw dataset comprised $1.44 \times 10^6$ 16S rRNA effective sequences and $1.44 \times 10^6$ ITS effective sequences. The mean length of these two sets of sequences was $252.5 \pm 0.5$ nt and $234.3 \pm 10.1$ nt, respectively. Based on the 97% similarity threshold, the sequences represented $7.74 \times 10^3$ and $2.15 \times 10^3$ OTUs, respectively. The 16S rRNA sequences originated from an average of $6.02 \times 10^4$, $5.67 \times 10^4$, $5.06 \times 10^4$, $4.49 \times 10^4$, and $2.97 \times 10^4$ OTUs at the level of phylum, class, order, family, and genus respectively, while the equivalent numbers for the ITS sequences were $4.04 \times 10^4$, $3.56 \times 10^4$, $3.56 \times 10^4$, $3.29 \times 10^4$, and $3.29 \times 10^4$ (Table 1). All raw sequences were deposited in NCBI (National Center of Biotechnology Information) under the accession number PRJNA525548.

**Table 1.** Operational taxonomic units (OTUs) and their abundance in the dazomet (DZ)- and biofungicide (BF)-treated soils.

| Microbe | Treatment | Phylum ($\times 10^4$) | Class ($\times 10^4$) | Order ($\times 10^4$) | Family ($\times 10^4$) | Genus ($\times 10^4$) |
|---|---|---|---|---|---|---|
| | Control-2015 | 6.16 ± 0.01 [a] | 5.88 ± 0.01 [a] | 5.38 ± 0.05 [a] | 4.94 ± 0.05 [a] | 3.54 ± 0.02 [a] |
| | DZ-2015 | 5.99 ± 0.07 [bc] | 5.52 ± 0.03 [c] | 4.71 ± 0.04 [d] | 4.13 ± 0.03 [d] | 2.72 ± 0.03 [d] |
| Bacteria | BF-2015 | 6.02 ± 0.01 [b] | 5.63 ± 0.03 [bc] | 5.06 ± 0.01 [c] | 4.40 ± 0.01 [c] | 3.01 ± 0.03 [bc] |
| | Control-2016 | 6.04 ± 0.01 [b] | 5.75 ± 0.07 [b] | 5.26 ± 0.05 [ab] | 4.72 ± 0.04 [b] | 3.21 ± 0.03 [b] |
| | DZ-2016 | 5.92 ± 0.03 [c] | 5.53 ± 0.04 [c] | 4.77 ± 0.04 [d] | 4.09 ± 0.04 [d] | 2.47 ± 0.02 [e] |
| | BF-2016 | 6.00 ± 0.02 [bc] | 5.71 ± 0.03 [b] | 5.16 ± 0.04 [bc] | 4.67 ± 0.04 [b] | 2.89 ± 0.03 [cd] |
| Average | | 6.02 | 5.67 | 5.06 | 4.49 | 2.97 |
| | Control-2015 | 4.76 ± 0.09 [a] | 4.30 ± 0.05 [b] | 4.30 ± 0.04 [b] | 4.04 ± 0.04 [a] | 4.01 ± 0.04 [a] |
| | DZ-2015 | 4.08 ± 0.18 [b] | 3.13 ± 0.12 [c] | 3.08 ± 0.13 [c] | 2.85 ± 0.07 [b] | 2.67 ± 0.21 [b] |
| Fungi | BF-2015 | 4.39 ± 0.22 [ab] | 3.09 ± 0.13 [c] | 3.12 ± 0.12 [c] | 2.96 ± 0.13 [b] | 2.91 ± 0.13 [b] |
| | Control-2016 | 4.58 ± 0.03 [a] | 4.70 ± 0.08 [a] | 4.70 ± 0.08 [a] | 4.08 ± 0.05 [a] | 4.00 ± 0.04 [a] |
| | DZ-2016 | 3.29 ± 0.11 [c] | 3.05 ± 0.14 [c] | 3.05 ± 0.14 [c] | 2.97 ± 0.16 [b] | 2.87 ± 0.16 [b] |
| | BF-2016 | 3.17 ± 0.05 [c] | 3.08 ± 0.07 [c] | 3.08 ± 0.07 [c] | 2.82 ± 0.07 [b] | 2.76 ± 0.08 [b] |
| Average | | 4.04 | 3.56 | 3.56 | 3.29 | 3.20 |

Data given in the form mean ± standard error. Control: no treatment, DZ: dazomet fumigation and BF: B. subtilis biofungicide. [a, b, c, d] and [e] indicated different levels of significant difference. Different letters associated with pairs of mean values indicate significant differences ($p < 0.05$) in microbial abundance between the soil treatments.

### 3.2. Alpha Diversity Analysis of the Soil Microbiome

Compared to the control treatment, the bacterial population in the BF-treated soil was more diverse in both years (6.1% more in 2015 and 4.2% more in 2016). There was no significant difference in fungal diversity in 2015, but there was a significant reduction (21.3%) in 2016 (Table 2). Compared

to the control treatment, the DZ treatment resulted in no significant differences in bacterial diversity in 2015 and a significant reduction (8.0%) in 2016 but did not affect fungal diversity in both years. The BF treatment resulted in a significantly higher bacteria species richness. The highest Faith's PD (phylogenetic diversity) values for the bacteria were associated with the 2015 sample of BF-treated soil, and the lowest with DZ-treated soils in both 2015 and 2016. The BF treatment induced the greatest bacterial evenness in both years, and the greatest fungal evenness in 2015.

**Table 2.** Bacterial and fungal alpha diversity indices in the treated soils.

| Soil Microbiota | Treatment | Diversity (Shannon) | Richness (Chao1) | Faith's PD | Evenness |
|---|---|---|---|---|---|
| Bacteria | Control-2015 | 9.625 ± 0.079 [c] | 4051.292 ± 28.873 [b] | 296.633 ± 1.903 [b] | 0.996 ± 0.000 [b] |
| | DZ-2015 | 9.704 ± 0.033 [c] | 3915.980 ± 22.684 [c] | 277.514 ± 1.730 [c] | 0.996 ± 0.000 [b] |
| | BF-2015 | 10.216 ± 0.053 [a] | 4518.900 ± 42.726 [a] | 320.292 ± 8.509 [a] | 0.998 ± 0.000 [a] |
| | Control-2016 | 9.577 ± 0.040 [c] | 3953.088 ± 28.553 [bc] | 295.582 ± 4.951 [b] | 0.995 ± 0.000 [c] |
| | DZ-2016 | 8.814 ± 0.025 [d] | 3170.464 ± 29.194 [d] | 230.511 ± 0.851 [d] | 0.992 ± 0.000 [d] |
| | BF-2016 | 9.979 ± 0.015 [b] | 4020.658 ± 9.112 [bc] | 294.688 ± 0.200 [b] | 0.997 ± 0.000 [a] |
| Fungi | Control-2015 | 6.000 ± 0.314 [ab] | 811.333 ± 29.525 [ab] | 184.107 ± 15.443 [a] | 0.952 ± 0.018 [a] |
| | DZ-2015 | 5.720 ± 0.271 [bc] | 751.791 ± 34.465 [bc] | 186.743 ± 6.017 [a] | 0.933 ± 0.010 [a] |
| | BF-2015 | 6.551 ± 0.115 [a] | 863.942 ± 11.664 [a] | 217.206 ± 5.406 [a] | 0.968 ± 0.003 [a] |
| | Control-2016 | 6.404 ± 0.080 [ab] | 787.092 ± 13.298 [ab] | 192.826 ± 5.303 [a] | 0.959 ± 0.003 [a] |
| | DZ-2016 | 6.229 ± 0.122 [ab] | 800.751 ± 18.828 [ab] | 183.855 ± 6.544 [a] | 0.963 ± 0.004 [a] |
| | BF-2016 | 5.041 ± 0.335 [c] | 702.647 ± 36.261 [c] | 192.406 ± 18.111 [a] | 0.882 ± 0.023 [b] |

Data given in the form mean ± SE. Control: no treatment, DZ: dazomet fumigation and BF: B. subtilis biofungicide. Faith's PD: phylogenetic diversity. [a, b, c, d] and [e] indicated different levels of significant difference. Different letters associated with pairs of mean values indicate significant differences (0.05) in microbial abundance between the soil treatments.

### 3.3. Beta Diversity Analysis of the Soil Microbiome

Non-metric multi-dimensional scaling analysis was used to identify differences in microbiome composition in both the DZ- and BF-treated soils (Figure 1). The analysis highlighted a distinct difference in the composition of the bacterial and fungal components in response to both treatments. On the basis of distance between the points, the control, BF, and DZ treatments had a distinct effect on the bacterial communities in both years, and a greater impact appeared in the second year compared with the first year (Figure 1a). The effect of the DZ treatment was distinct from that of both the BF and the control treatments with respect to the first MDS (multi-dimensional scaling) in both years. For the fungal component of the microbiome, the DZ treatment had an effect which differentiated it from the control and BF in both years (Figure 1b).

Weighted UniFrac and unweighted UniFrac distances were used to describe the variation in the beta diversity of the DZ- and BF-treated soils (Figure 2a). Based on the Weighted UniFrac distances, the level of beta diversity in the bacterial population was 0.37 (2015) and 0.51 (2016) between the control and DZ treatments, and 0.28 (2015) and 0.37 (2016) between the control and BF treatments. In the fungal population, the differences between the control and DZ treatments were 1.28 and 1.38 in 2015 and 2016, respectively (Figure 2b). The differences between the control and BF treatments were 0.80 and 0.95 in 2015 and 2016, respectively. Similarly, the differences between the control and DZ treatments were much larger than between the control and the BF treatments for both the bacterial and fungal population. According to the unweighted UniFrac distances, the bacterial beta diversity was 0.35 between the control and DZ treatments in 2015 and 0.42 in 2016. The equivalent values for the comparison between the control and BF treatments were 0.31 in 2015 and 0.34 in 2016 (Figure 2a). The fungal beta diversities were 0.45 and 0.57 between the control and DZ treatments in 2015 and 2016, respectively (Figure 2b). The fungal beta diversities were 0.45 and 0.44 between the control and BF treatments in 2015 and 2016, respectively. Similarly, the estimated beta diversity within the fungal component was larger between the control and DZ treatments than between the control and BF treatments in 2016.

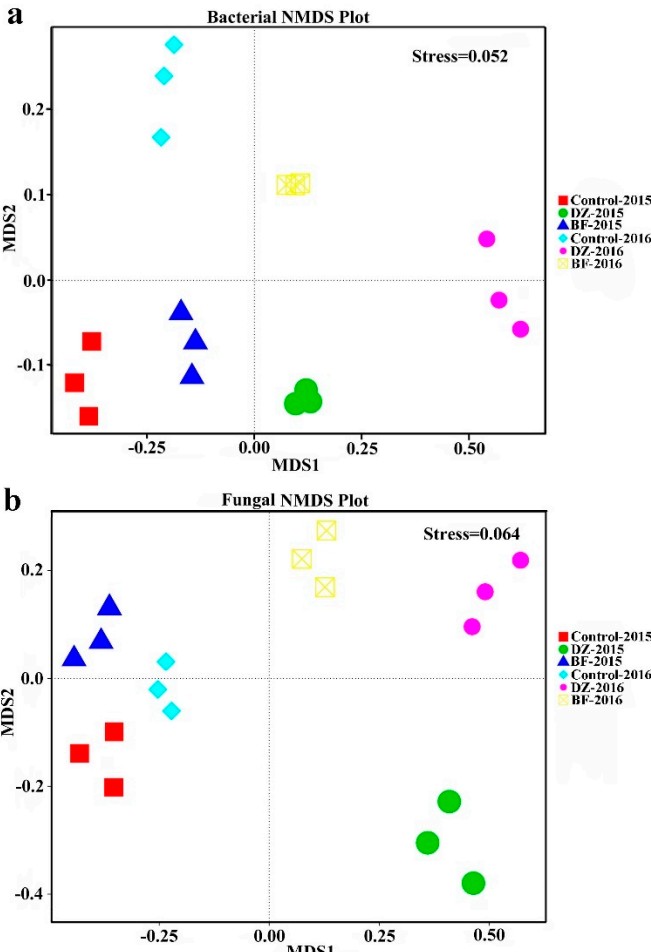

**Figure 1.** Non-metric multi-dimensional scaling analysis was used to demonstrate the effect of soil treatment over two years on the composition of the soil microbiome. (**a**) Bacterial component, (**b**) Fungal component. Control: no treatment, DZ: dazomet fumigation and BF: *B. subtilis* biofungicide. Number 2015 and 2016 indicate sampling time.

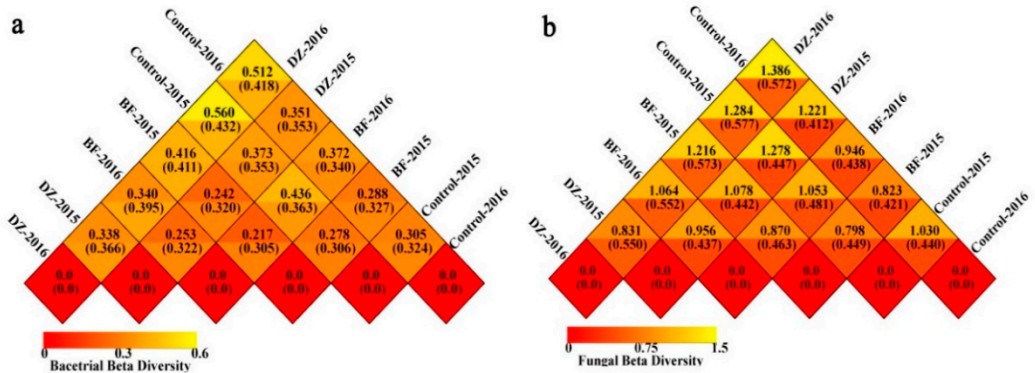

**Figure 2.** Beta diversity analysis illustrating the effect of soil treatment over two years on the composition of the soil microbiome. The heat map shows the effect on (**a**) the bacterial component, (**b**) the fungal component. The numbers shown outside the parentheses refer to the UniFrac-weighted analysis, and those inside the parentheses refer to the UniFrac-unweighted analysis. Number 2015 and 2016 indicate sampling time.

### 3.4. Taxonomic Composition of the Microbiome

The ten most abundant phyla present in the rhizosphere samples following the various treatments are presented in Figure 3. The most highly represented bacterial phyla belonged to the *Proteobacteria*, *Gemmatimonadetes*, *Firmicutes*, *Actinobacteria*, *Bacteroidetes*, *Chloroflexi*, *Acidobacteria*, *Thaumarrchaeota*, and *Planctomycetes*, together accounting for >90% of the bacterial community (Figure 3a). The ten most highly represented fungal phyla were *Ascomycota*, *Mortierellomycota*, *Chytridiomycota*, *Basidiomycota*, *Aphelidiomycota*, *Olpidiomycota*, *Rozellomycota*, *Glomeromycota*, *Mucoromycota,* and *Calcarisporiellomycota*, which together accounted for >50% of the fungal community (Figure 3b). The different treatments showed similar phylum compositions but differed in terms of the relative abundance of various groups.

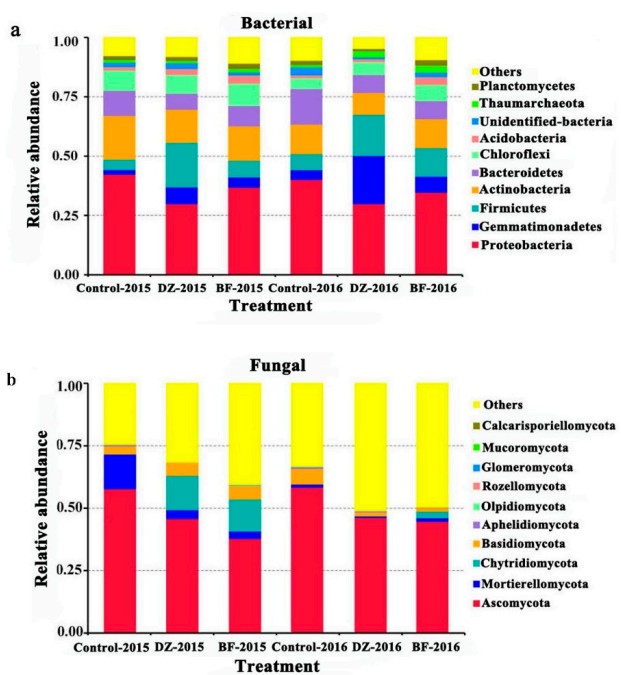

**Figure 3.** The relative abundance in treated soil of microbes by phylum. (**a**) Bacterial phylum, (**b**) Fungal phylum. Control: no treatment, DZ: dazomet fumigation and BF: *B. subtilis* biofungicide. Number 2015 and 2016 indicate sampling time.

In 2015, the DZ and BF treatments significantly lowered the abundance of both *Proteobacteria* and *Bacteroidetes* (Supplementary Table S1). The DZ-treated soil contained the greatest abundance of *Gemmatimonadetes* and *Firmicutes* when compared with control and BF, and the BF treatment contained the greatest *Gemmatimonadetes* compared with control and DZ. Meanwhile, the DZ treatment decreased the relative abundance of *Ascomycota, Chytridiomycota,* and *Rozellomycota*, while it promoted the presence of *Aphelidiomycota* and *Mucoromycota*. The BF treatment also suppressed the growth of *Ascomycota*, while promoting that of *Chytridiomycota* and *Glomeromycota*.

In 2016, the DZ and BF treatments significantly lowered the abundance of both *Proteobacteria* and *Bacteroidetes* (Supplementary Table S1). For the DZ-treated soil, there was a marked reduction in the representation of *Proteobacteria, Actinobacteria*, and *Planctomycetes*, and the BF treatment enhanced the representation of *Actinobacteria, Chloroflexi, Acidobacteria,* and *Thaumarchaeota*. Meanwhile, the DZ-treated soil decreased the relative abundance of *Ascomycota, Chytridiomycota,* and *Rozellomycota*, and in the BF-treated soil, *Basidiomycota* featured strongly and *Mucoromycota* featured weakly.

A heat map analysis of the abundant genus within a hierarchical cluster based on Bray–Curtis distance indices showed different patterns of community structure among the different treatments (Figure 4). In 2015, the most abundant bacterial genera were *Flavobacterium, Sphingopyxis,* and *Pesudomonas* in the control (Figure 4a). However, the DZ and BF treatments significantly increased

the abundance of unidentified *Acidobacteria* and *Bacillus* in both years when compared with the control. Among the fungi, species belonging to the genera *Plectospharella*, *Issatchenkia*, *Rhodotorula*, *Dactylonectria,* and *Mortierella* were the most frequently encountered in the control (Figure 4b). The DZ and BF treatments increased the abundance of *Trichoderma*, while they decreased the abundance of *Fusarium* compared to the control. In 2016, the most highly represented bacterial genera in the control were *Glycomyces, Mariniflexile and Devosia*. And the most abundant fungal genera were *Chaetomium*, *Colletotrichum* and *Penicillium*. The BF treatment decreased the abundance of *Fusarium* and increased the abundance of *Trichoderma*, while, the DZ treatment had the opposite trend with BF treatment.

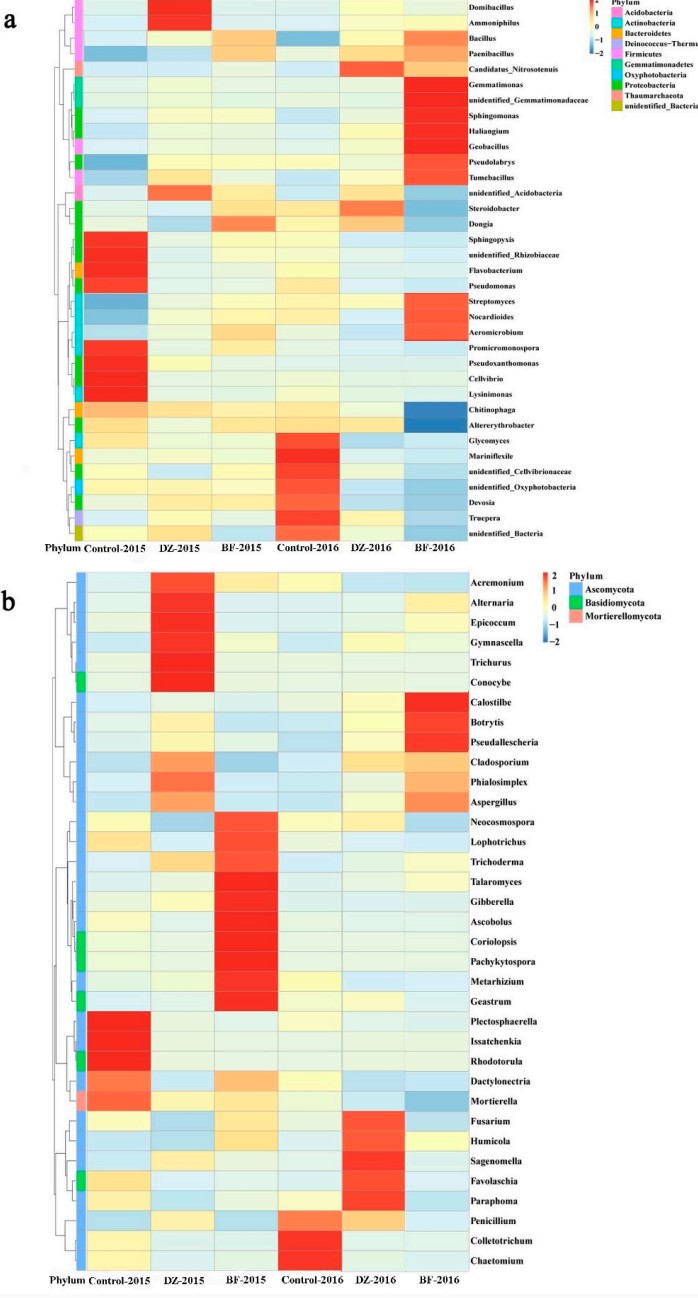

**Figure 4.** Heat map analysis at genus level in different soil treatments (**a**) Bacterial genera, (**b**) Fungal genera. Control: no treatment, DZ: dazomet fumigation and BF: *B. subtilis* biofungicide. Number 2015 and 2016 indicate sampling time.

The effect of the DZ and BF treatments on the abundance of *F.oxysporum* is illustrated in Figure 5a. In the 2015 samples, both treatments strongly suppressed this taxon, reducing its abundance by 84.6% and 75.9%, respectively. In the 2016 samples, however, the DZ treatment induced a significant ($p <$ 0.05) increase (43.8%) in the abundance of the pathogen compared to the control, while the abundance of *F.oxysporum* remained effectively suppressed by the BF treatment. The effect of the treatments on the abundance of *Bacillus* spp. is shown in Figure 5b: the greatest abundance of this taxon was recorded in the BF-treated soil in both years. Compared to the control, the relative abundance of *Trichoderma* spp. differed between the DZ and BF treatments in both years (Figure 5c). In 2015, this taxon was about 11.7-fold more abundant in both treated soils than in the control soil, while in 2016, its greatest abundance was associated with the BF treatment, reaching a level 4.3-fold greater than in either the control or the DZ-treated soils. According to a linear regression analysis, the relative abundance of both *Bacillus* spp. ($R^2$ = 0.77, $p$ = 0.02) and *Trichoderma* spp. ($R^2$ = 0.83, $p$ = 0.03) was significantly negatively associated with the abundance of *F.oxysporum* (Figure 5d,e).

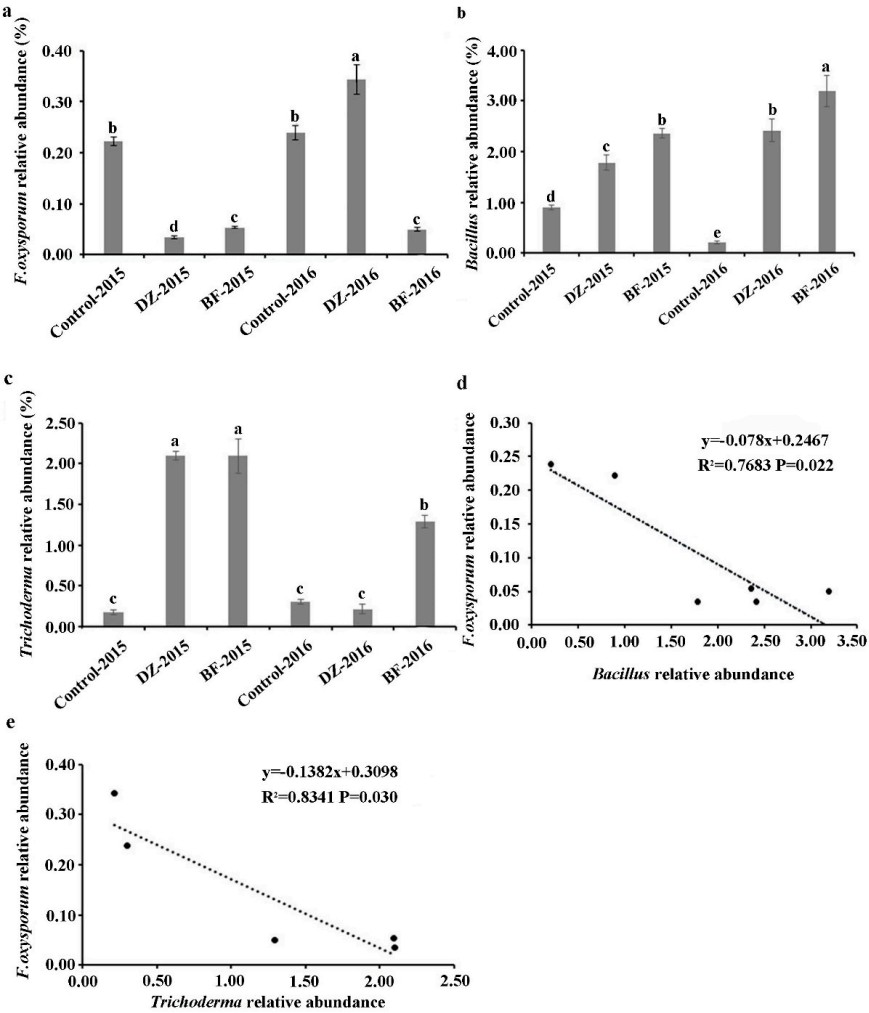

**Figure 5.** The relative abundance of key soil microbiome taxa in treated soil. (**a**) *F. oxysporum*, (**b**) *Bacillus* spp., (**c**) *Trichoderma* spp. (**d,e**) A linear regression analysis illustrating the correlation of the abundance of *F. oxysporum* on that of (**d**) *Bacillus* spp., (**e**) *Trichoderma spp*. Control: no treatment, DZ: dazomet fumigation and BF: *B. subtilis* biofungicide. Number 2015 and 2016 indicate sampling time. Letters above the bars indicate a significant difference according to Duncan's multiple range test at the $p$ < 0.05 level.

## 4. Discussion

The composition and diversity of the rhizosphere microbiome exert a substantial influence over plant and soil health [21], although optimal conditions can be specific to plant species [22,23], soil type [24], and crop management practice [25,26]. Here, we characterized the effect on the rhizosphere microbiome associated with chrysanthemum plants of either fumigating with a commonly used compound or providing *Bacillus subtilis* as a biofungicide. The experiments confirmed the observations reported by Feld et al. [27], which showed that the bacteria was significantly compromised by DZ fumigation, which is doubtless a direct consequence of the toxin on a range of soil microbe taxa. In contrast, in response to the BF treatment, the diversity of the bacterial component was found to increase over time, while that of the fungal component continued to fall, as has similarly been shown by You et al. [28]. This positive response reflects the colonization by *B. subtilis* of the rhizosphere, which promotes the growth of other plant-associated bacterial taxa [7,28]. At the same time, the biofilm generated by *B. subtilis* acts antagonistically on the development of fungal taxa [29], thereby contributing to the suppression of soil-borne fungal pathogens [30].

The composition of the bacterial and fungal communities in chrysanthemum rhizosphere were differentially affected by the two soil treatments, as has been noted repeatedly in other systems [28,31–33]. The nature of the soil treatment played a large part in shaping the composition of the rhizosphere microbiome. Among the bacterial phyla identified, the *Proteobacteria* proved to be the most abundant, irrespective of the soil treatment. The prevalence of members of this taxon has been positively correlated with the occurrence of the disease take-all in wheat, caused by the fungus *Gaeumannomyces graminis var. tritici* [34]. Both the DZ and BF treatments suppressed *Proteobacteria* to some degree in both the sample years. Certain species (*Streptomyces, Nocardioides, Aeromicrobium*) belonging to the phylum *Actinobacteria* have been considered as potential antibacterial agents [35] and their presence has been linked to disease suppression [36]. Here, the BF-treated soil proved to harbor the highest abundance of *Actinobacteria* in phylum and the highest abundance of *Streptomyces, Nocardioides,* and *Aeromicrobium* at the genera level in 2016. A large number of pathogenic fungal species are members of phylum *Ascomycota*, such as the *Fusarium* genera [37]. This group of species was notably less abundant in both the DZ-treated in 2015, and particularly in the BF-treated soils in 2016, consistent with their general depletion in disease-suppressive soils [38], an effect which has been attributed to the inhibition imposed by *B. subtilis* on the growth of various *Ascomycete* pathogens [3,39].

Among the bacterial genera identified, the DZ and BF treatments significantly increased the abundance of unidentified *Acidobacteria* and *Bacillus* in both years, and these two genera were associated with disease suppression [10,14]. *Bacillus* can form a stable and extensive biofilm and secrete many antifungal compounds that protect plants against attack by soil-borne pathogens [3]. *Acidobacteria* might be involved in the biogeochemical cycles of the rhizosphere soil and improve the resistance of plants [40]. The enrichment of these genera was probably due to the increase in available niches after soil fumigation or fungicide [3]. Among the fungi, the BF treatments increased the abundance of *Trichoderma, Metarhizium,* and *Mortierella*. *Trichoderma* are plant growth-promoting fungi that enhance plant nutrient uptake, production of growth hormones, and protect plants from pathogen infection [41]. *Mortierella* are known to compete with pathogens for resources and produce antibiotics to suppress pathogens [6], and *Metarhizium* are entomopathogenic fungi, which can be used to control harmful insects on plants [42].

*Fusarium oxysporum* includes damaging pathogen formae speciales in many continuous cropping systems [3]. The present data show that fumigation with DZ was successful in suppressing *F. oxysporum* over the course of the first year, but its effectiveness was reduced in the subsequent year. An explanation for this inconsistency could be due to the disordered soil community structure and diversity after soil fumigant DZ application. In contrast, the BF treatment showed similar effectivity in both years, 2015 and 2016, which is consistent with the negative correlation which has been established between the abundance of *B. subtilis* and *F. oxysporum* in the rhizosphere [30]. The treatment also boosted the population of *Trichoderma spp.*, with the result that their abundance was similarly significantly

negatively correlated with that of *F. oxysporum*. Certain members of this genus have been shown to express antifungal activity [9], so these taxa may have also contributed to the control of *F. oxysporum* in the BF-treated soil.

## 5. Conclusions

In summary, both the BF and DZ treatments altered the composition of the rhizosphere microbiome developed around the roots of chrysanthemum. The former treatment increased the abundance of *B. subtilis* and *Trichoderma*, which can include strains likely provided some control by suppressing the growth of *F. oxysporum*. Our results provided theoretical guidance for the control of *Fusarium* wilt of chrysanthemum and other ornamental plants, improved the knowledge of the composition of fungal communities in the rhizosphere soil after chemical fungicide and bio-fungicide application, and it can lead to a better understanding of fungal roles in these soil ecosystems.

**Supplementary Materials:** The following are available online at http://www.mdpi.com/2077-0472/9/12/255/s1, Table S1: The relative abundance in the treated soil of the ten most frequently occurring bacterial and fungal phyla.

**Author Contributions:** Conceptualization, F.C., W.F. and S.Z.; Data curation, H.C.; Formal analysis, H.C.; Funding acquisition, S.C., Z.G., F.C., W.F. and S.Z.; Investigation, J.Z. and J.J.; Methodology, J.Z. and J.J.; Project administration, S.C., Z.G., F.C., W.F. and S.Z.; Supervision, S.C. and Z.G.; Writing—original draft, H.C.; Writing—review and editing, H.C. and S.Z.

**Funding:** This research was supported by Fund for Independent Innovation of Agricultural Sciences in Jiangsu Province (CX(16)2020), Project for Agricultural Technology Extension Service Pilot from Scientific Research Institute in Jiangsu Province (TG(17)002), the Fundamental Research Funds for the Central Universities (KYZ201833), the Policy Guidance Program of Jiangsu Province (BY2016077-06), The program for key research and development, Jiangsu, China (BE2019384) and a project Funded by the Priority Academic Program Development of Jiangsu Higher Education Institutions.

**Acknowledgments:** The authors thank Fengge Zhang for offering technical support.

**Conflicts of Interest:** The authors declare no conflict of interest.

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
