# Peer review of "Assessing the Influence of Fumigation and Bacillus Subtilis-Based Biofungicide on the Microbiome of Chrysanthemum Rhizosphere"

_agriculture, doi:10.3390/agriculture9120255_

Round 1

Reviewer 1 Report

The changes made in this submission addressed the reviewer's comments from the first submission.

Author Response

Reply to Reviewer 1

Manuscript Number: agriculture-643296

Title: Assessing the Influence of Fumigation and Bacillus Subtilis-based Biofungicide on the Microbiome of Chrysanthemum Rhizosphere

Authors: Huijie Chen, Jiamiao Zhao, Jing Jiang, Sumei Chen, Zhiyong Guan, Fadi Chen, Weimin Fang and Shuang Zhao *

Dear respectful reviewer,

Thanks again for your review on our manuscript. We are also much grateful to you and other reviewers for putting forward the valuable suggestions in our manuscript, which are very helpful for the revision of our manuscript. 

Best wishes,

Yours sincerely,

Shuang Zhao

Shuang Zhao, Ph.D.

On behalf of all the authors

Reviewer 2 Report

I do not feel that the authors sufficiently addressed my concerns. They did add some genus-level analysis, but hardly go on to discuss it. They should have repeated their measures of diversity with the genus level analysis. I still feel like this work could be published, but the analysis just is not biologically sound in my opinion.

Author Response

Reply to Reviewer 2

Manuscript Number: agriculture-643296

Title: Assessing the Influence of Fumigation and Bacillus Subtilis-based Biofungicide on the Microbiome of Chrysanthemum Rhizosphere

Authors: Huijie Chen, Jiamiao Zhao, Jing Jiang, Sumei Chen, Zhiyong Guan, Fadi Chen, Weimin Fang and Shuang Zhao *

Dear respectful reviewer,

Thanks for your comments on our manuscript. We are also much grateful to you and other reviewers for putting forward the valuable suggestions in our manuscript, which are very helpful for the revision of our manuscript. In the past days, we have made great efforts to carefully revise the manuscript according to the valuable comments and suggestions of you and another reviewer. They all have been taken into account in the revised manuscript and all of the revisions were using the “Track Changes” function in Microsoft Word. Below we present the point-by-point responses to your comments.

Best wishes,

Yours sincerely,

Shuang Zhao

Shuang Zhao, Ph.D.

On behalf of all the authors

Comment 1: I do not feel that the authors sufficiently addressed my concerns. They did add some genus-level analysis, but hardly go on to discuss it. They should have repeated their measures of diversity with the genus level analysis.

Answer: Thanks for your suggestion. We have added the discussion about the structural composition of microbial communities at the genus level. Please see the line 318-329 in the new revision.

Reviewer 3 Report

The aim is in line with the scope of the Journal.

Author Response

Reply to Reviewer 3

Manuscript Number: agriculture-643296

Title: Assessing the Influence of Fumigation and Bacillus Subtilis-based Biofungicide on the Microbiome of Chrysanthemum Rhizosphere

Authors: Huijie Chen, Jiamiao Zhao, Jing Jiang, Sumei Chen, Zhiyong Guan, Fadi Chen, Weimin Fang and Shuang Zhao *

Dear respectful reviewer,

Thanks again for your positive comments on our manuscript. We are also much grateful to you and other reviewers for putting forward the valuable suggestions in our manuscript, which are very helpful for the revision of our manuscript. 

Best wishes,

Yours sincerely,

Shuang Zhao

Shuang Zhao, Ph.D.

On behalf of all the authors

Round 2

Reviewer 2 Report

I feel that this manuscript could be stronger based on the data that were generated.

This manuscript is a resubmission of an earlier submission. The following is a list of the peer review reports and author responses from that submission.

Round 1

Reviewer 1 Report

The experimental design and the statistical analyses appear sound. Please check the grammar in lines 28 and 302-304. IN lines 203-204, mention the values for the beta-diversity. Increase the font sizes in Figures 3 and 4.

In this work, the authors attempted to characterize the correlation between the bacterial and fungal diversity in soil and the application of specific chemical and biological fungicide treatment. The goal was to generate a quantifiable knowledge of soil biodiversity upon biocide treatment, which would then inform the cultivation process of commercially important flower Chrysanthemum. Considering the economic impact, this study is relevant and interesting. While the study was very simple in experimental design and did not involve a number of environmental and artificial factors related to biodiversity in soil, it addresses a gap in the knowledge regarding the effect of fungicide application in soil. The manuscript text is sound and the conclusions appears to be supportive of the authors’ hypotheses. 

Author Response

Reply to Reviewer 1

Manuscript Number: agriculture-591667

Title: Assessing the Influence of Fumigation and Bacillus Subtilis-based Biofungicide on the Microbiome of Chrysanthemum Rhizosphere

Authors: Huijie Chen, Jiamiao Zhao, Jing Jiang, Sumei Chen, Zhiyong Guan, Fadi Chen, Weimin Fang and Shuang Zhao *

Dear respectful reviewer,

Thank you very much for your positive comments on our manuscript. We are also much grateful to you and other reviewers for putting forward the valuable suggestions in our manuscript, which are very helpful for the revision of our manuscript. In the past days, we have made great efforts to carefully revise the manuscript according to the valuable comments and suggestions of you and another reviewer. They all have been taken into account in the revised manuscript and the revised parts were marked in red in Microsoft Word. Below we present the point-by-point responses to your comments.

Best wishes,

Yours sincerely,

Shuang Zhao

Shuang Zhao, Ph.D.

On behalf of all the authors

Comment 1: Please check the grammar in line 28 and 302-304.

Answer: Thanks for your suggestion. The grammar in line 28 and 302-304 were checked. Please see the line 28 and 321-323 in the new revision.

Comment 2: In lines 203-204, mention the values for the beta-diversity.

Answer: Thank you for your suggestion. The values for the beta-diversity were added in the new revision. Please see line 196-200 and 203-205.

Comment 3: Increase the font sizes in Figures 3 and 4.

Answer: Thank you for your suggestion. The two pictures were revised. Please see the line 238 (Figure 3) and 275 (Figure 5) in the new revision.

Reviewer 2 Report

The authors in this study compared treatment of chrysanthemum with DZ and BF, to reduce fungal infection. They found that BF was more consistent in preventing infection and more consistent in determining soil microbial diversity. Overall, despite a low number of replicates, I approve of the experimental design. However, the analysis is greatly lacking. There are many instances where the methods need to be explained more thoroughly, especially in the statistical analysis and filtering of OTUs. Additionally, I would recommend that the authors use ASVs instead of OTUs since they allow for clearer comparisons across studies. Another major shortcoming of the analysis is that the authors only identified taxa to the phylum level. This does not allow for any meaningful conclusions about the ecology of the organisms in the samples. I strongly recommend that the analysis in this paper be repeated. Below are my more detailed comments. line 37 - change fumigate to fumigating line 95 - change The 9 plots experiment was to The nine experimental plots were line 139 - It is now more acceptable to use ASV’s instead of OTU’s. I would recommend that the authors consider redoing their analysis with ASVs but I do not think it will dramatically change the results Lines 130-146 - Did you filter to remove chloroplast OTUs? Line 172 - These data would be better presented with box plots Line 213 - How was this determined? Why types of taxa were excluded? Lines 211-260 - Phylum is a very course taxonomic group. I imagine that if the authors looked at order or genus, they would see much greater shifts in both bacterial and fungal community composition. The 16S and ITS sequences have enough resolution to go to genus level. 
 Lines 211-260 - The methods for these statistics to conclusions about significant differences in this section are not described. Also, there is no description of how the authors calculated relative abundance. Lines 256-260 - It seems that for some target organisms the authors chose to look at genus level. Please describe these methods and do this overall. Lines 271-274 - It is not necessary to speculate on this point. You should be able to look into your data to see if the relative abundance of Bacillus has increased. Lines 275-289 - All this discussion regarding taxa at the phylum level is too speculative. If the authors had looked at more refined taxonomic levels (which they should have been able to do with their data) they could make more concrete conclusions.

Author Response

Reply to Reviewer 2

Manuscript Number: agriculture-591667

Title: Assessing the Influence of Fumigation and Bacillus Subtilis-based Biofungicide on the Microbiome of Chrysanthemum Rhizosphere

Authors: Huijie Chen, Jiamiao Zhao, Jing Jiang, Sumei Chen, Zhiyong Guan, Fadi Chen, Weimin Fang and Shuang Zhao *

Dear respectful reviewer,

Thanks for your comments on our manuscript. We are also much grateful to you and other reviewers for putting forward the valuable suggestions in our manuscript, which are very helpful for the revision of our manuscript. In the past days, we have made great efforts to carefully revise the manuscript according to the valuable comments and suggestions of you and another reviewer. They all have been taken into account in the revised manuscript and the revised parts were marked in red in Microsoft Word. Below we present the point-by-point responses to your comments.

Best wishes,

Yours sincerely,

Shuang Zhao

Shuang Zhao, Ph.D.

On behalf of all the authors

Comment 1: The analysis is greatly lacking. There are many instances where the methods need to be explained more thoroughly, especially in the statistical analysis and filtering of OTUs.

Answer: We have reorganized the methods of Illumina sequencing and statistical analysis in the revised manuscript according to your comment (line 123-147).

Comment 2: I would recommend that the authors use ASVs instead of OTUs since they allow for clearer comparisons across studies.

Answer: Thanks for your suggestion. Your suggestion is pretty good, However, the ASVs you proposed is relatively new at present in China. And the biotechnology sequencing companies and experts are still designing relevant procedures and algorithms. So far, OTU is still used in most researches. And we will learn to use ASVs to analyze the sequencing data in our future research.

Comment 3: Another major shortcoming of the analysis is that the authors only identified taxa to the phylum level. This does not allow for any meaningful conclusions about the ecology of the organisms in the samples. I strongly recommend that the analysis in this paper be repeated.

Answer: Thanks for your suggestion, the taxa to the genus level was added in the new revision. Please see line 242-258.

Comment 4: line 37 - change fumigate to fumigating.

Answer: Thanks for your suggestion. We have changed "fumigate" to "fumigating". Please see line 37 in the new revision.

Comment 5: line 95 - change The 9 plots experiment was to The nine experimental plots were.

Answer: Thanks for your suggestion. We have changed "The 9 plots experiment was" to "The nine experimental plots were". Please see line 94 in the new revision.

Comment 6: line 139 - It is now more acceptable to use ASV’s instead of OTU’s. I would recommend that the authors consider redoing their analysis with ASVs but I do not think it will dramatically change the results Lines.

Answer: Thanks for your suggestion. And we have known that using amplicon sequence variants (ASVs) to analyze the high-throughput sequencing data would be more accurate. However, the biotechnology sequencing companies and experts are still designing relevant procedures and algorithms in China. And we will learn to use ASVs to analyze the sequencing data in our future research.

Comment 7: 130-146 - Did you filter to remove chloroplast OTUs?

Answer: No, we didn’t filter to remove chloroplast OTUs. Because the abundance of chloroplast OTUs was very low, and the influence on the results of the sequencing was very little.

Comment 8: Line 172 - These data would be better presented with box plots.

Answer: Thanks for your suggestion. According to your suggestion, we tried to display these data with box plots, but we found that eight box diagrams were needed in the picture to present these data, so the content of picture would be very small. Thus, we suggest to use the table to present these data. In addition, the data presented by table is also accepted and adopted by many published papers (Wu et al., 2017).

(Wu, X., Guo, S., Jousset, A., Zhao, Q., Wu, H., Rong, L., Kowalchuk, G. A. and Shen, Q. (2017) Bio-fertilizer application induces soil suppressiveness against Fusarium wilt disease by reshaping the soil microbiome. SOIL BIOL BIOCHEM 114, 238-247.)

Comment 9: Line 213 - How was this determined? Why types of taxa were excluded?

Answer: Based on our study, the dominant bacterial and fungal phyla in soil ecosystems refers to the high level of abundance. The relative abundance of phyla less than 0.02 were excluded. And the line 213 was revised. Please see line 215-216 in the new revision.

Comment 10: Lines 211-260 - Phylum is a very course taxonomic group. I imagine that if the authors looked at order or genus, they would see much greater shifts in both bacterial and fungal community composition. The 16S and ITS sequences have enough resolution to go to genus level.

Answer: Thanks for your suggestion. We added the result of bacterial and fungal community composition in the genus level in the new revision, please see the line in 242-258.

Comment 11: Lines 211-260 - The methods for these statistics to conclusions about significant differences in this section are not described. Also, there is no description of how the authors calculated relative abundance.

Answer: Thanks for your suggestion. We added the description of the methods about significant differences analysis in the new revision, please see the line in 261 and 277. In our study, the relative abundance was calculated by the abundance of the effective tags corresponding to taxon/ the abundance of total effective tags.

Comment 12: Lines 256-260 - It seems that for some target organisms the authors chose to look at genus level. Please describe these methods and do this overall.

Answer: Thanks for your suggestion. This part was revised. Please see line 241-257 in the new revision.

Comment 13: Lines 271-274 - It is not necessary to speculate on this point. You should be able to look into your data to see if the relative abundance of Bacillus has increased.

Answer: Thanks for your suggestion. The lines 271-274 were revised. Please see line 288-289 in the new revision.

Comment 14: Lines 275-289 - All this discussion regarding taxa at the phylum level is too speculative.

Answer: Thanks for your suggestion. This part was revised. Please see line 292-307 in the new revision.

Reviewer 3 Report

In this experiment, the effects of DZ fumigation and the use of B. subtilis-based biofungicide on the microbiome associated with chrysanthemum roots were studied.

The studies were carried out correctly in terms of methodology. They bring new knowledge to soil microbiology. I suggest that you make some minor changes in the manuscript.

On lines 149-155 and the first line of Table 1, use a hard space instead of the *. With such low values of PCA1 and PCA2, Figure 1 is not reliable. I suggest giving up of this Figure and the text in lines 194-204. Latin names of taxa (lines 215-240, 278, 281285, 286,289) and species / generic (lines 245, 246, 248,252,253, 254, 256-259, 326). I suggest writing in italics.

Author Response

Reply to Reviewer 3

Manuscript Number: agriculture-591667

Title: Assessing the Influence of Fumigation and Bacillus Subtilis-based Biofungicide on the Microbiome of Chrysanthemum Rhizosphere

Authors: Huijie Chen, Jiamiao Zhao, Jing Jiang, Sumei Chen, Zhiyong Guan, Fadi Chen, Weimin Fang and Shuang Zhao *

Dear respectful reviewer,

Thank you very much for your positive comments on our manuscript. We are also much grateful to you and other reviewers for putting forward the valuable suggestions in our manuscript, which are very helpful for the revision of our manuscript. In the past days, we have made great efforts to carefully revise the manuscript according to the valuable comments and suggestions of you and another reviewer. They all have been taken into account in the revised manuscript and the revised parts were marked in red in Microsoft Word. Below we present the point-by-point responses to your comments.

Best wishes,

Yours sincerely,

Shuang Zhao

Shuang Zhao, Ph.D.

On behalf of all the authors

Comment 1: On lines 149-155 and the first line of Table 1, use a hard space instead of the *.

Answer: Thanks for your suggestion. We have changed the “*” to “×” in the new revision. Please see line 150-158 in the new revision.

Comment 2: With such low values of PCA1 and PCA2, Figure 1 is not reliable. I suggest giving up of this Figure and the text.

Answer: Thanks for your suggestion. Because of the low values of PCA1 and PCA2, the principal component analysis (PCA) in the Figure 1 was abandoned. And considering the completeness of manuscript structure, we replaced the PCA with the Non-metric multi-dimensional scaling (NMDS) analysis in the Figure 1. Please see line 178-191 in the new revision.

Comment 3: in lines 194-204. Latin names of taxa (lines 215-240, 278, 281285, 286,289) and species / generic (lines 245, 246, 248,252,253, 254, 256-259, 326). I suggest writing in italics.

Answer: We have revised the latin names of taxa and species/generic in italics in the revised manuscript according to your comment (line 217-221, 225-236, 243-252, 258, 262-263, 265, 270, 295, 299-300, and line 302-304).